

# Voxel inversion of airborne electromagnetic data for improved groundwater model construction and prediction accuracy

**N. K. Christensen[1], and T.P.A Ferre[2], G. Fiandaca[1], S. Christensen[1]**

[1]{Department of Geoscience, Aarhus University, Aarhus,Denmark}

[2]{Department of Hydrology and Water Resources, University of Arizona, Tucson, USA.}

Correspondence to: N. K. Christensen (phda.nikolaj.kruse@geo.au.dk)

## Abstract

We present a workflow for automated construction and calibration of large-scale groundwater models that includes the integration of airborne electromagnetic (AEM) data and hydrological data. The AEM data are inverted to form a 3D geophysical model. The parameter of interest is the hydraulic conductivity, which can be determined by translating the 3D geophysical model, using a petrophysical relationship, to form a 3D hydraulic conductivity distribution. We use the geophysical models and hydrological data to determine the optimum spatially distributed petrophysical relationship. The two shape factors of the petrophysical relationship primarily work as translators between resistivity and hydraulic conductivity, but the shape factors can also compensate for structural defects in the model.

The method is demonstrated for a synthetic case study. The AEM data are inverted with both smoothness (smooth) and minimum gradient support (sharp) constraints, resulting in two competitive geophysical models. The value of the AEM data quality is tested by inverting the alternative geophysical models using data corrupted with four different levels of background noise. Subsequently, the geophysical models are used to construct two competing groundwater models. The performance of the flow model was tested for four types of prediction. All predictions occurred beyond the calibration base. Predictions of a pumping well's recharge area and groundwater age were applying the same stress situation as applied during hydrologic model calibration, while predictions of head and stream discharge was done for a stress situation changed from those applied during hydrologic model calibration.

The results show that geophysical models inverted with sharp constraints improve the predictive capability of the groundwater models compared to geophysical models inverted with smooth constraints. It was found that the use of sharp models improves the prediction of recharge area, while





prediction of groundwater age does not improve significantly. When the stress situation is changed the
prediction of head change and stream discharge improves significantly for sharp models compared to
smooth models. This is especially true for predictions of head change made in the vicinity of the
pumping well and far-away from hydrologic boundaries. Furthermore, the geophysical data quality has
variable influence on different model predictions. Prediction accuracy improves with AEM data
quality for predictions of recharge area, head change and stream discharge, while the accuracy appears
to not improve for prediction of groundwater age.
## 1  Introduction
Large-scale geological and groundwater models are used extensively to support aquifer management.
(Here "large scale" refers to an area of from tens to thousands of square kilometers.) Determining the
distribution of hydraulic properties and the geometry and connectivity of the groundwater system is of
significant importance since these features control the flow paths (Desbarats and Srivastava 1991;
Fogg et al. 1999; Weissmann and Fogg 1999). Incorrect reconstruction of the geological structures has
thus been recognized as the most important source of uncertainty when a groundwater model is used to
make predictions outside its calibration base (Refsgaard et al. 2012; Seifert et al. 2012; Zhou et al.
2014). The data traditionally used for structural mapping include lithological logs from boreholes,
hydrological data, and hydraulic testing results, but these data are often sparse and uneven distributed
within an investigated domain. In these (very common) cases, data scarcity becomes a major obstacle
for structural mapping in relation to large scale groundwater modeling (Refsgaard et al. 2012; Zhou et
al. 2014).
Ground-based and airborne electromagnetic method (AEM) methods have shown a great potential for
mapping of geological structures (Jørgensen et al. 2003; Thomsen et al. 2004; Abraham et al. 2012;
Oldenborger et al. 2013; He et al. 2014; Munday et al. 2015). For large scale mapping AEM is an
efficient and cost-effective method by which the traditional data can be supplemented by dense
estimates of electrical resistivity which in some environments inform about the lithology and thereby
about structure (Robinson et al. 2008; Binley et al. 2015). The AEM measurements can quickly be
made over large areas, and the resolution can be as fine as 25 m in the horizontal direction and 5 m in
the vertical (Schamper et al. 2014) with penetration depth of up to several hundred meters (Siemon et
al. 2009).
Various methods have been reported for how to incorporate resistivity estimates (from now on called
resistivity models) in groundwater model construction. Manual and knowledge driven approaches
have been used to combine geological, hydrological and geophysical data with expert knowledge
(Jørgensen et al. 2013). However, the manual approach is subjective and possibly very time





consuming and expensive to use when resistivity models from large AEM surveys are to be
incorporated in model construction. Alternatively, more objective and cost-efficient geostatistical
modeling approaches (Carle and Fogg 1996; Deutsch and Journel 1998; Strebelle 2002) are available
for generating models from a combination of borehole information and AEM determined resistivity
models. For example: He et al. (2014) used a transition probability indicator simulation approach
(Carle and Fogg 1996), while Gunnink and Siemon (2015) used sequential indicator simulation
(Deutsch 2006). Marker et al. (2015) used a deterministic strategy for the integration of AEM
resistivity models into the hydrological modeling process.
The just mentioned studies all used sequential hydrogeophysical inversion approaches (SHI; as
defined by Ferré et al. 2009). In SHI the geophysical data are inverted first and independently from the
later inversion of the hydrological data. For large scale groundwater modeling Herckenrath et al.
(2013) and Christensen et al. (2016) were using both SHI and joint hydrogeophysical inversion
approaches (JHI; as defined by Ferré et al. 2009). By JHI the geophysical and hydrological data are
inverted jointly by linking the geophysical and hydrological models directly through some of their
parameters. The linking can for example be done by using an Archie's law inspired petrophysical
relationship (Archie 1942) to translate between the geophysical and hydrologic parameters.
In general, petrophysical relationships are difficult to establish, because such translation tend to be
site, scale and facies specific (Chen et al. 2001; Hyndman and Tronicke 2005; Slater 2007) and
uncertain (Mazáč et al. 1985; Slater 2007). The studies by Herckenrath et al. (2013) and Christensen et
al. (2016) were using a fixed petrophysical relationship throughout the model domain. Better results
can potentially be obtained by using a spatially variable relationship which allows for local translation
between hydraulic conductivity and electrical resistivity, and by including the spatially dependent
petrophysical parameters in the optimization process (Linde et al. 2006).
There are two other challenges for incorporating resistivity models into large scale groundwater
modeling: difference in model discretization, and choice of geophysical regularization methodology.
Groundwater models are often discretized in a regular voxel grid while the traditional resistivity
models are 1D and placed at the respective sounding location. For airborne surveys, for example, the
resistivity models are normally located along the flight lines (Christiansen et al. 2006). Such resistivity
models therefore need to be relocated to conform to the grid of the groundwater model. The relocation
will often be a subtle process where information easily can be lost. To accommodate this issue
Fiandaca et al. (2015) presented a geophysical modeling approach referred to as "voxel inversion"
which decouples the geophysical inversion model space from the geophysical measurement positions.
This allows estimation of a 3D geophysical model that is discretized on the same voxel grid as the
groundwater model.



Traditionally, geophysical regularization includes horizontal and vertical smoothing constrains
(Constable et al. 1987) or a few layer inversion (Auken and Christiansen 2004), whereas a
groundwater system often has sharp layer or body boundaries. It has therefore been recognized, e.g. by
Day-Lewis (2005) and others, that the regularization used to stabilize the geophysical inversion may
not reflect the actual hydrologic conditions unless it is chosen carefully. If, for example, smooth
regularization is used to estimate resistivity models in a sharply layered system it will produce a
blurred resistivity distribution from which one should be careful with inferring the spatial distribution
of hydraulic conductivity to be used in a groundwater model. In this case it would be better to use
minimum gradient support regularization (Portniaguine and Zhdanov 1999; Blaschek et al. 2008;
Vignoli et al. 2015) for the geophysical inversion because thus estimated resistivity distribution tend to
consist of fewer and more sharply defined layer boundaries (vertically and horizontally).
In this paper we present a sequential hydrogeophysical approach for using a voxel based 3D resistivity
model to parameterize and calibrate a groundwater model. We will demonstrate that the model
parameterization allows the calibration to compensate for errors in the resistivity model. We will also
demonstrate that it is important for groundwater modeling that the underlying resistivity model is
estimated by regularization constraints that conform well to the geological environment. Finally we
analyze how groundwater model prediction accuracy depends on the quality of the geophysical data
that was used to estimate the resistivity model. Section 2 of the paper presents the methodology.
Section 3 describes the synthetic test case used for our demonstration purposes. Section 4 presents the
results, while sections 5 and 6 discusses and draws the conclusions, respectively.





## 2 Methodology

Conceptually, the methodology defines a translator function that describes the petrophysical relationship between electrical resistivity and hydraulic conductivity. A fundamental aspect is that the petrophysical relationship can vary horizontally and vertically, thereby adapting to the local conditions in translation from the geophysical model space to the hydrological model space. Through inversion, the 3D spatially dependent optimal parameters of the petrophysical relationship are estimated for each layer interval, thereby covering the entire three-dimensional model space.

Figure 1 provides a workflow for the method. First, the gathered airborne electromagnetic (AEM) data from the survey area are inverted with smooth or sharp horizontal and vertical constraints (Vignoli et al. 2015). This is done by using a recently developed voxel inversion scheme which decouples the geophysical model from the position of the acquired data (Fiandaca et al. 2015). The geophysical model space thus corresponds to the full 3D hydrological model grid. Secondly, the geophysical voxel based resistivity model is used as input for the subsequential hydrological inversion. The geophysical model parameter (resistivity) is linked to the main investigated parameter (hydraulic conductivity) through a petrophysical relationship which has unknown shape factor values. The shape factor values are estimated through a hydrological inversion which minimizes an objective function describing the misfit between simulated groundwater model responses and corresponding observed hydrological data. Finally, the calibrated groundwater model can be used to make a set of relevant hydrologic predictions. The various steps of the methodology are explained in more detail in the following.

### 2.1 Geophysical voxel inversion

The AEM data undergoes constrained deterministic inversion (Figure 1, box 1) using a recently developed voxel inversion approaches. This approach allows the geophysical model spaces to be spatially decoupled from the geophysical measurement positions (Fiandaca et al. 2015). In most inversion schemes the forward and inverse formulations use the same model discretization for both inversion and forward calculation, but in the voxel formulation the two model discretizations are decoupled. The voxel model space thus defines the geophysical properties on a set of nodes of a regular 3D grid.

For calculating the forward responses, a "virtual" 1D model is a built at each sounding position. The "virtual" 1D model is defined by a number of layers, and layer thicknesses. The geophysical properties are interpolated from the voxel model space into the layer centers of the virtual model that is subsequently used to simulate the forward response for the corresponding sounding.





The voxel inversion approach thus allows for inverting AEM data into a geophysical model defined on
a 3D regular grid, regardless of the sounding positions. This implies that the geophysical inversion can
be conducted using the same grid as that defined for a 3D groundwater model. Scaling issues in the
coupling of geophysical and hydrological models can thus be avoided by using the same spatial
discretization.
The general solution to the non-linear geophysical inversion problem can be found in Auken et al.
(2014). To stabilize the inverse problem, either of two types of regularization methods can be applied.
The first regularization method is commonly referred to as smoothness-constrained inversion
(Constable et al. 1987). The smoothness-constrained inversion tends to reduce contrasts and the
resulting geophysical model may appear blurred. The reason for this is found in its minimum-structure
L2 norm inversion formalism (Constable et al. 1987; Menke 2012), which following the notation used
by Vignoli et al. (2015) can be expressed as:

$$\left(m_i - m_j\right)^2 / {\sigma_{i,j}}^2$$

$$( 1 )$$

where the $m_i$ and $m_j$ are the constrained parameters and $\sigma_{i,j}$ defines the constraint strength. The
penalization of structures is clearly seen in eq. ( 1 ), where $\left(m_i - m_j\right)_k^2 / {\sigma_{i,j}}^2$ is proportional to the
square of the value of the variation $(m_i - m_j)$. This implies that an increase in model parameter
variation will always result in a penalization in the stabilizer. The smoothness regularization thus
prevents reconstruction of sharp transitions.
The second regularization method is the minimum gradient support (Portniaguine and Zhdanov 1999;
Blaschek et al. 2008; Vignoli et al. 2015), which allows for large sharp vertical and horizontal model
transitions. The minimum gradient support regularization seeks to minimize the spatial variations
vertically and laterally by penalizing the vertical and horizontal model gradients through the stabilizer
expressed as (Vignoli et al., (2015)):

$$\frac{\left(m_i - m_j\right)^2 / {\sigma_{i,j}}^2}{\left(m_i - m_j\right)^2 / {\sigma_{i,j}}^2 + 1}$$

$$( 2 )$$





In eq. ( 2 ) $\sigma_{i,j}$ is a parameter used to control the sharpness of the regularization constraints. The
stabilizer contribution to the objective function is thus one when $|m_i - m_j| \gg \sigma_{i,j}$ and zero when
$\sigma_{i,j} \gg |m_i - m_j|$. The minimum gradient support functional thus counts the number of model
variations larger than $\sigma_{i,j}$ for the stabilizer term of the objective function. This formalism thus allows
sharp vertical and horizontal model transitions, which are excessively penalized by the smoothness-
constrained inversion.

## 2.2  Hydrological model parametrization

Section 2.1 describes an inversion methodology for which the geophysical property distribution can be
estimated for each element in a voxel grid (Figure 1, box 1). The three dimensional distribution of
electrical resistivity values is linked to the main investigated hydrological parameters (e.g. hydraulic
conductivity) through a petrophysical relationship.
Linking hydraulic conductivity and electrical resistivity is not trivial because the parameter values and
the form of the petrophysical relationship may vary dramatically between different types of
environments.  In addition, there can be fundamental questions about how the effective properties
controlling electrical current flow are related to the effective properties controlling fluid flow (Slater
2007). The primary factors controlling this relationship are porosity, pore water conductivity,
tortuosity, grain size, degree of saturation, amount of clay minerals, etc. (McNeill 1980). The simplest
petrophysical relationship is the empirical relationship known as Archie´s law (Archie 1942) that
relates porosity, pore water conductivity, and the degree of saturation to bulk electrical conductivity.
However, this type of relationship does not take the electrical surface conductance on the surface of
clay minerals into account. The Waxman and Smith model (Waxman and Smits 1968) combined with
the dual-water model by Clavier et al. (1984) provides a basis for establishing empirical relationships
for shaly sand and sediments containing clays (Revil and Cathles 1999; Revil et al. 2012). For glacial
sedimentary environments, it is reported that clay has low electrical resistivity and also low hydraulic
conductivity, and sand has high electrical resistivity and high hydraulic conductivity (Mazáč et al.
1985). It is common to use a power law relationship which is given some theoretical support by
Purvance and Andricevic (2000). The relationship is expressed as

$$K = \alpha \cdot \rho^{\beta} \qquad\qquad (3)$$

where $K$ is the hydraulic conductivity (m/s), $\rho$ is the electrical resistivity (ohm-m), and $\alpha$ and $\beta$ are
two empirical shape factors. To compute $K$ for each element in the groundwater model grid, $\alpha$ and $\beta$





need to be parameterized and estimated. We suggest to make the parameterization by pilot points
placed in a regular grid in each layer of the groundwater model (Certes and De Marsily 1991; Doherty
2003). Each pilot point holds a set of $\alpha$ and $\beta$ parameters, and kriging is used for spatial interpolation
of $\alpha$ and $\beta$ from the pilot points to the model grid. This kind of parametrization creates smooth
transitions in the parameter fields and allows for variation in both the horizontal and vertical direction
of the $\rho$ to K translation. Hydraulic conductivity can thus be calculated by eq. ( 3 ) for every element
in the groundwater model grid.
## 2.3   Hydrological Inversion
The model parameters, $\alpha$ and $\beta$ at the pilot points, are calibrated by fitting the groundwater model to
hydrological data. When the number of model parameters is large compared to the number of
observation data, the minimization must be stabilized by regularization.  The total objective function
to be minimized is therefore a balanced compromise between a measurement term ($\Phi_m$)  and a
regularization term ($\Phi_r$). The combined objective function has the form

$$\Phi_{total} = \Phi_m + \mu \cdot \Phi_r = \sum_{i=1}^{n_d} \omega_{d,i}\left(d_{obs,i} - d_{sim,i}\right)^2 + \mu \cdot \Phi_r \qquad ( 4 )$$

where $\Phi_{total}$ is the total objective function,  $d_{obs,i}$  and $d_{sim,i}$  are measured and equivalent simulated
data values, $\omega_{di}$ is a data dependent weight, $\mu$ is a weight factor, and $\phi_r$ is a Tikhonov regularization
term. Here, $\phi_r$ is defined as preferred difference regularization, where the preferred difference
between neighboring parameter values is set to zero. $\Phi_{total}$ is minimized iteratively, and the
regularization weight factor, $\mu$, is calculated during the iteration in a way so $\Phi_m$, the measurement part
of the objective function, becomes approximately equal to a user specified target value (Doherty

22   2010).

# 3   Synthetic example
For illustrative purposes we use a three dimensional synthetic system very similar to that presented by
Christensen et al. (2015). The only difference is that the active part of the groundwater system only
consists of 5 layers whereas Christensen et al. (2015) used a 20 layer model.





## 3.1 Groundwater reference system and hydrological data

The groundwater system is intended to mimic a glacial landscape and covers an area that is 7000 m (N-S) by 5000 m (E-W). The geology of the system was generated using T-PROGS (Carle 1999) as having a horizontal discretization of 25 m x 25 m, and a vertical discretization of 10 m. The system extends 50 m in the vertical direction where it reaches impermeable clay with a horizontal surface. The T-PROGS generated geology above the impermeable clay consists of categorical deposits of sand, silt and clay. Within each of the three types of deposits, hydraulic conductivity, recharge and the porosity were generated as horizontally correlated random fields using FIELDGEN (Doherty 2010). All boundaries of the domain were defined as having no-flow conditions except the southern boundary where hydraulic head was defined as constant, h = 0 m. The local recharge depends on the type of sediment at the uppermost layer. Most groundwater discharges through the southern boundary, but approximately 35% discharges into a river running north to south in the middle of the domain (Figure 2). Groundwater flow was simulated as confined steady-state flow employing MODFLOW-2000 (Harbaugh et al. 2000) with the spatial discretization equal to the geological discretization. Groundwater is pumped at a rate of 0.015 $m^3s^{-1}$ from a well located at x=2487.5m and y=1912.5 m and the well screens the deepest 10 meters of the groundwater system. In the following this system is called the *reference system*.

Thirty-five boreholes are found within the domain (Figure 2). Each borehole contains a monitoring well that screens the deepest 10 m of sand registered in the borehole. For each system realization, hydraulic head in the 35 wells and the river discharge at the southern boundary were extracted from a forward simulation made by MODFLOW-2000. The 35 simulated hydraulic head values were contaminated by independent Gaussian error with zero mean and 0.1 m standard deviation. The river discharge was corrupted with independent Gaussian error with zero mean and a standard deviation corresponding to 10 % of the true river discharge. The 36 contaminated values constitute the hydrological data used for groundwater model calibration.

## 3.2 Geophysical reference system and data

The geophysical reference system was designed so there is perfect correlation between hydraulic conductivity and electrical resistivity. This implies that a relationship between hydraulic conductivity and measured electrical resistivity is likely to exist. The true relationship is of the same form as eq. ( 3 ), and it uses constant shape factor values $\alpha = 1e^{-12}$ and $\beta = 4$. This corresponds to conditions where clay has low electrical resistivity and also low hydraulic conductivity, and sand has high





electrical resistivity and high hydraulic conductivity. The impermeable clay at the base of the
reference system was assigned a constant value of 5 ohm-m.
The AEM data were simulated using AarhusInv (Auken et al. 2014) for a system setup similar to a
typical dual-moment SkyTEM-304 system (Sørensen and Auken 2004). The simulated survey consists
of 35 E-W flight lines with 200 meter spacing between the flight lines. AEM system responses were
simulated for every 25 m along the flight lines giving a total of 6300 sounding locations for both the
transmitted high and low moments. AarhusInv is a 1D modeling code. To mimic the loss of resolution
with layer depth we simulated the responses using the 2D logarithmic average resistivity of all model
cells inside the radius of the foot print at a given depth.. To obtain the geophysical data set, the
simulated data were contaminated with noise according to the noise model suggested by (Auken et al.

11  2008):

$$V_{resp} = V \cdot \left( 1 + G(0,1) \cdot \left[ STD^2{}_{uni} + \left( \frac{V_{noise}}{V} \right)^2 \right]^{1/2} \right) \qquad (5)$$

where $V_{resp}$ is the perturbed synthetic data, $V$ is the synthetic noiseless data, $G(0,1)$ is standard
Gaussian noise (with zero mean and unit standard deviation), and $STD^2{}_{uni}$ is uniform noise variance.
$V_{noise}$ is the background noise contribution given by

$$V_{noise} = b \cdot \left( \frac{t}{10^{-3}} \right)^{-1/2}, \qquad (6)$$

where $t$ is the gate center time in seconds, and $b$ is the background noise level at 1 ms. For the
following analysis we generated geophysical datasets with four levels of background noise, i.e. $b$ equal
to 1, 3, 5, and 10 nV/m$^2$, respectively. The uniform standard deviation, which accounts for instrument
and other non-specified noise contributions, was set to 3% for d**B**/dt responses. After the data were
perturbed with noise, it was processed as a field data set (Auken et al. 2009), resulting in an uneven
number of gates per sounding. Figure 3 illustrates the resulting low and high moment AEM sounding
data, respectively, for the different background noise levels.





### 3.3 Geophysical voxel inversion

The geophysical data were inverted by voxel inversion (Fiandaca et al. 2015) using AarhusInv (Auken et al. 2014). The voxel inversion was conducted in two different ways: by using L2-norm "smooth" constraints, or by using minimum gradient support "sharp" constraints (both implemented in AarhusInv; Auken et al. 2014).

To avoid the influence of numerical discretization errors, the geophysical voxel inversion uses the same spatial discretization as the reference system and the groundwater model. For both smooth and sharp inversions a 40 ohm-m uniform half-space was used as starting model, and spatial regularization was applied using the same settings throughout all inversions. It was unnecessary to apply vertical constraints for any of the inversions. (On the contrary, depth and direction dependent horizontal constraint factors were used for both smooth and sharp inversions. For smooth regularization constraint factors of 1.9 along the flight lines and 1.05 perpendicular to the flight lines was used for the first layer. The constraint factors was set to decrease linear with depth, resulting in constraint factors of 1.4 along the flight lines and 1.02 perpendicular to the flight lines for the sixth layer. For sharp inversion, constraint factors of 1.0625 along the flight lines and 1.01 perpendicular to the flight lines was used for the first layer, while factors of 1.025 along the flight lines and 1.01 perpendicular to the flight lines was used for the sixth layer.

### 3.4 Groundwater model parametrization and calibration

In the following the groundwater model will be parameterized in two different ways. Both ways treat the shape factors $\alpha$ and $\beta$ of the relationship (3) between hydraulic conductivity and resistivity as spatially dependent parameters to be estimated. The two parameterizations differ by the resistivity model that is used to calculate the hydraulic conductivity field of the groundwater model:

- The first type of parameterization uses a resistivity model estimated by smooth voxel inversion of AEM data collected with a background noise level of 3 nV/m$^2$. These models will be referred to as SHI-smooth-3.
- The second type of parameterization usesa resistivity model estimated by sharp voxel inversion of AEM data collected with a background noise level of either 1, 3, 5, or 10 nV/m$^2$. These models will be referred to as SHI-sharp-1, SHI-sharp-3, SHI-sharp-5, and SHI-sharp-10, respectively.

The shape factors $\alpha$ and $\beta$ of the petrophysical relationship are parametrized by placing pilot points in a uniform grid, with 5 nodes in the x direction and 7 in the y direction. Hence, in total the groundwater model is parameterized by 5x7x5 = 175 petrophysical relationships each having two parameters (the shape factors).




The parameter values are estimated by fitting the available hydrological data consisting of the 35
observations of hydraulic head and one river discharge observation. Calibration is done by
minimization the total objective function given by eq. ( 4 ), where the measurement objective function
is computed as

$$\Phi_m = n_h{}^{-1} \sum_{i=1}^{n_h} \omega_h \big(h_{obs,i} - h_{sim,i}\big)^2 + n_r{}^{-1} \sum_{i=1}^{n_r} \omega_r \big(r_{obs,i} - r_{sim,i}\big)^2 \qquad (7)$$

where, $n_h$ and $n_r$ are the number of head and river measurements, respectively; $h_{obs}$ and $h_{sim}$ are
observed and corresponding simulated hydraulic heads; $r_{obs}$ and $r_{sim}$ are observed and
corresponding simulated river discharge; and $\omega_h$ and $\omega_r$ are subjectively chosen weights for head and
discharge data, respectively. We strived at choosing values of $\omega_h$ and $\omega_r$ that give an average value
of $\bar{\phi}_m = 2$, where the average is calculated over the 20 system realizations. Such values of $\omega_h$ and $\omega_r$
will then be estimates of the reciprocal of the total error variance for hydraulic head and discharge,
respectively. This is seen from (7). By trial and error we found $\omega_h = 1$ and $\omega_r = 1.38 \cdot 10^5$ which
gave $\bar{\phi}_m = 2.5$. The value $\omega_h = 1$ corresponds to $(10 \cdot \sigma_h)^{-2}$ where $\sigma_h$ is the standard deviation for
the head measurements. The value $\omega_r = 1.38 \cdot 10^5$ corresponds to $(\sigma_r)^{-2}$, where $\sigma_r$ is the standard
deviation of the measurement error for measured discharge. These values thus indicate that the
calibrated models have error in their simulation of hydraulic head but not in simulation of river
discharge.
Calibration was performed using local search as optimization implemented in the parameter estimation
software BeoPEST, a version of PEST (Doherty 2010) that allows the inversion to run in parallel
using multiple cores and computers.
It should be noted that for calibration and model prediction we applied the recharge field and boundary
conditions of the reference system.
**3.5   Reference and model predictions**
The calibrated SHI-smooth and SHI-sharp groundwater models are evaluated by comparing their
simulated model predictions with corresponding predictions simulated for the reference system. The
former are called "model predictions, the latter are called "reference predictions".
Prediction types 1 and 2 relate to steady-state flow when groundwater is pumped from the well. This is
also the condition for which the hydrologic data used for calibration were sampled. Type 1 is the





average age of the groundwater pumped from the well. Type 2 is the size of the recharge area of the
pumping well. Both these predictions differ in type from the calibration data. For these model
predictions we used a homogeneous porosity of 0.2 (the average value of the reference system porosity
fields is 0.184).
Prediction types 3 and 4 relate to a new stress situation long after pumping from the well has ceased:
type 3 is groundwater discharge into the stream, and type 4 is head recovery for a well screening layer
north-east of the pumping well (location is shown in Figure 2).
The reference and model prediction types 3 and 4 were simulated by MODFLOW-2000 (Harbaugh et
al. 2000), while type 1 and 2 were simulated by forward particle tracking using MODPATH version 5
(Pollock 1994) and MODFLOW-2000 results.
The first two types of prediction are interesting from the perspectives of protection and resource-
management of a well field, while the latter two are relevant in the case of possible change of
management practice resulting in a new stress.

## 3.6  Evaluation of prediction performance

As said in the beginning of section 2, steps 1-3 of the framework can be repeated for a number of
system realizations for making consistent statistical interference on the model prediction results. Here
20 different reference system realizations were used. For each prediction we hereby have 20
corresponding sets of reference predictions and model predictions that can be used to evaluate the
performance of a calibrated model with respect to that prediction. The performance is evaluated for
SHI-smooth and SHI-sharp models, respectively, and it is done in the following ways.
Prediction error characteristics are quantified by the mean absolute error ($MAE$), the mean error ($ME$),
respectively:

$$MAE = \frac{1}{N} \sum_{i=1}^{N} |x_i - t_i| \qquad (8)$$

$$ME = \frac{1}{N} \sum_{i=1}^{N} x_i - t_i \qquad (9)$$





where $x_i$ is the model prediction of realization $i$, $t_i$ is the reference prediction of realization $i$, and
$N = 20$ is the number of system realizations. $MAE$ measures how close the model prediction tends to
be to the reference prediction; $ME$ measures the tendency of positive or negative bias in the model
prediction.

## 6   4   Results

### 7   4.1   Geophysical results

Figure 4 shows a representative cross-section for one of the 20 system realizations. Both geophysical
models in Figure 4 were inverted using data perturbed with a background noise level of $3nV/m^2$.
Comparing the geophysical model results with the reference model, we find that the SHI-smooth-3
resolves the main features reasonably well for the upper layers. The main discrepancy is found in the
fifth layer where the sand bodies are not resolved.  In general, the resistivity of the sand bodies (dark
orange in the reference system) is underestimated and the transitions between the categorical deposits
are artificially smooth.
Figure 4 shows that SHI-sharp-3 resolves the sand body in layer 5 much better than SHI-smooth-3.
Moreover, the locations and boundaries of the geological deposits tend to be less smeared out when
using the sharp constraints. Inspection of the histograms at the bottom of Figure 4 shows that the SHI-
sharp-3 model tends to produce resistivity distributions that have more similarities with the reference
distributions than the SHI-smooth-3 model. This improvement should potentially allow for easier
translation from electrical resistivity into hydraulic conductivity and more faithful representation of
hydrogeologic structure and connectivity.
Figure 5 shows corresponding voxel by voxel density plots of reference versus estimated electrical
resistivity for a SHI-smooth model and corresponding SHI-sharp models. Pearson's correlation
coefficient (PCC; Cooley and Naff 1990) is shown on top of the density plot for each layer. A
comparison of the density plots and the PCC values of the SHI-smooth-3 and SHI-sharp-3 models
shows that using sharp instead of smooth constraints improves the inverted geophysical model. The
improvement is most clearly seen for the sand deposits
For both SHI-smooth and SHI-sharp models there is a strong correlation between the electrical
resistivity estimates and the true electrical resistivities of the first layer, but the SHI-smooth model has
weaker correlation than the SHI-sharp models. For both type of models the correlation weakens with
depth and background noise. The former is caused by the resolution limitations of AEM data.




However, the depth and resistivity of the low-resistivity clay at the base of the model are well resolved
by both the SHI-smooth and SHI-sharp models inversions (results not shown).
## 4.2  Hydrological calibration results
The calibration results for the 20 different system realizations are shown in Figure 6. The figure shows
the measurement objective function value, $\Phi_m$, for each system realization. We aimed at using
weights that would make the minimized measurement objective function value averaged over the 20
system realizations approximately equal to 2. Figure 6 shows that this is nearly satisfied by the SHI-
Sharp models even for large background noise levels. For many of the realizations the SHI-Smooth
model also fits the data well, but for a couple of realizations the misfit is much larger than aimed at.
This makes $E[\Phi_m]$ equal to 5.8 for SHI-Smooth-3 models while it is 2.5for the SHI-Sharp-3 models.
This indicates that the estimated hydraulic conductivity field tends to be less wrong for sharp models
than for smooth models.
## 4.3  Parameter estimation
Figure 7 shows a cross section of the estimated $K$-, $\alpha$- and $\beta$- fields for one of the system realizations .
The two columns show estimates for the SHI-smooth-3 and  SHI-sharp-3 models, respectively. Figure
8 shows a density plot of the reference hydraulic conductivity distribution and the estimated hydraulic
conductivity distributions. The results in Figure 7 and Figure 8 are typical for all 20 system
realizations.
From Figure 7 a) and Figure 7 b) it is seen that the estimated $\alpha$ and $\beta$ parameter values are changing
smoothly in the horizontal direction but have sharp transitions in the vertical direction. The second
row of Figure 7 shows the corresponding estimated $K$ fields whose main features are determined by
the underlying resistivity models (Figure 4), but they are "corrected" during model calibration to make
the groundwater fit the hydrological data.
For the SHI-smooth-3 model, $\alpha$ and $\beta$ are taking compensatory roles particularly in the first layer.
Here the estimated $\alpha$ and $\beta$ values in this layer are higher than the shape factors of the true
relationship that was used to construct the geophysical reference system. This increases the hydraulic
conductivity in layer 1 to compensate for the too low hydraulic conductivity (and resistivity, Figure 4)
in layer 2 and deeper layers. The estimated $\alpha$ and $\beta$ values are not sufficient to compensate for the
missing deep high-resistivity body in in layer 5 of the SHI-Smooth-3 model (Figure 4).





For the SHI-sharp-3 model, the estimated $\alpha$ and $\beta$ parameter values only vary slightly from the shape
factor values of the true relationship except for layer 5 (Figure 7 b)). This indicates that for the more
shallow layers the sharp inversion of AEM data sufficiently resolves the resistivity of features that are
important for groundwater model calibration. In layer 5 the estimate of shape factor $\beta$ turns out to be
fairly high, this compensates for the too low resistivity estimates in this layer (Figure 4).
Figure 8 shows voxel by voxel density plots of reference versus estimated hydraulic conductivity for
SHI-smooth and SHI-sharp models. The figure is equivalent to Figure 5. Figure 8 confirms that the
resulting $K$ field tends to be overestimated for the first layer, and in particular for the SHI-smooth-3
model. From the second layer and down the hydraulic conductivity values tend to be underestimated
for sand but overestimated for silt and clay. Moreover, the distributions of estimated $K$ smear out with
depth. Judged by PCC values and visual inspection of Figure 8, the hydraulic conductivity field
estimated for SHI-sharp models is in better agreement with the reference field than the field estimated
by the SHI-Smooth-3 model.
Model structural accuracy is quantified in Table 1 for both the SHI-smooth and SHI-sharp models.
Structural accuracy is here calculated as the fraction of total number of voxels for which the estimated
$\log_{10}$-hydraulic conductivity plus/minus twenty percent contains the true $\log_{10}$-hydraulic conductivity
value of the reference model. The results are averaged over the 20 system realizations. From Table 1 it
is seen that all SHI-sharp models outperform the accuracy of the SHI-smooth models except for layer
5. The exception occurs because the SHI-smooth models are fairly good at estimating the $K$
distributions for silt and clays, but underestimates $K$ for sand (Figure 8). On the contrary, SHI-models
overestimate the $K$ distributions for silt and clays, but only slightly underestimate $K$ for sand (Figure
8). Therefore, for layer 5 the model structural accuracy appears to be better for SHI-smooth than for
SHI-sharp models.

## 4.4   Prediction results

For each of the 20 system realizations, the calibrated groundwater models were used to make the
model predictions described in section 3.5. Figure 9 shows scatter plots of reference prediction versus
the calibrated model prediction; each plotted point corresponds to a particular system realization and
corresponding SHI-smooth-3 or SHI-sharp-3 model. The mean error (*ME*) and mean absolute error
(*MAE*) of the prediction are also given in Figure 9. Figure 10 shows a *MAE* contour map for head
recovery predictions.





### 4.4.1   Particle tracking predictions
The first column of Figure 9 shows results for prediction of average age of the groundwater pumped
from the pumping well. The scatter plot illustrates that SHI-sharp models tend to over-predict average
age. This is seen by the majority of points plotting above the identity line as well as by the value of
$ME = 32$ (Figure 9). The age prediction results are similar for the SHI-smooth models although the
spread of points is larger than for SHI-sharp-3 (e.g. quantified by the larger value of $MAE$). There are
two major explanations for these relatively "poor" predictive performances. First, the calibrated K-
fields underestimate hydraulic conductivity of sand deposits in the deeper layers (Figure 8), which
results in too slow particle travel times at depth. Secondly, the reconstruction of the deepest layers is
too smooth for both SHI-smooth and SHI-sharp models (Figure 7) and does not resolve the small-scale
variability that controls the transport of particles.
The second column of Figure 9 is for prediction of the recharge area of the pumping well. The scatter
plot shows that the SHI-smooth models under-predicts the recharge area. This happens because the
smooth models lead to estimation of hydraulic conductivities in the deepest layers that are too low.
This creates a deep cone of depression around the pumping well that extends upward locally to reach
the river bed. This induces a local discharge of water from the stream through the groundwater system
to the pumping well. These models thus predict that a significant proportion of the pumping comes
from local discharge from the river. (This is compensated by increased model predicted groundwater
discharge to other parts of the river.) For the corresponding reference systems, the river is not losing
water, and all water pumped from the well origins from groundwater recharge.
The SHI-sharp models are better predictors of the recharge area, but also these models tend to predict
an area too small. These models also predict local discharge from the river to the groundwater system,
but to a lesser degree than the SHI-smooth models. This is likely because the main features of the
reference system are better reconstructed by the SHI-sharp-3 models.
### 4.4.2   Head recovery and discharge predictions
The prediction of head recovery at the observation well is done poorly by the SHI-smooth-3 (Figure
9). The predicted head recovery is very small for most of these models because they tend to have too
little hydraulic connectivity between the deepest layers, the estimated hydraulic conductivities are too
low in the deep sand layers, and the simulated cone of depression is therefore too deep and too local.
The SHI-sharp-3 models make less biased, fairly reasonable predictions of the head recovery (Figure
9) because they resolve the variations of hydraulic conductivity at depth better than the SHI-smooth-3
models. The superiority of  SHI-smooth-3 models for recovery prediction is also seen from the $MAE$





contour maps in Figure 9. The *MAE* is seen to be spatially dependent: it is largest at the pumping well,
and smallest at the constant head boundary to the south
The fourth column of Figure 9 shows that both types of models are good predictors of discharge to the
river after cessation of pumping. However, the SHI-sharp-3 model prediction is superior since its
points tend to plot close to the identity line. For SHI-smooth-3, the prediction tends to be positively
biased and more spread than for SHI-sharp-3.
### 4.4.3    Prediction error as function of data quality
In Figure 11   *MAE* is used as a metric to evaluate how the prediction performance of SHI-sharp
models depends on the level of background noise for the geophysical data. The noise levels were kept
unchanged for the hydrological data.
Figure 11 shows that the average age prediction made by SHI-sharp models are nearly unaffected by
the quality of the geophysical data. It is speculative, but this result may be because this prediction is
highly dependable on small scale variability in hydraulic conductivity and porosity that cannot be
resolved from any of the geophysical data sets.
It is different for the recharge area prediction (Figure 11): MAE increases for this by approximately
25% when the level of background noise is increased from 1 nV/m$^2$ to 10 nV/m$^2$. This happens
because the variations of resistivity (and thus hydraulic conductivity) are less well resolved from the
geophysical data of poor quality.
The third and fourth row of Figure 11 shows the head recovery and river discharge prediction after
cessation of the pumping well. Head recovery and discharge predictions also tend to depend on the
quality of the geophysical data. The *MAE* increases by 17 % for recovery prediction and 23 % for
discharge prediction when the noise level of the geophysical data increases from 1 nV/m$^2$ to 10 nV/m$^2$.



**5  Discussion**
**5.1  Estimation of Parameters in the Petrophysical Relation**
Parameterizing the groundwater model by assuming a spatially dependent petrophysical relationship
between resistivity and hydraulic conductivity makes it possible to use a resistivity voxel model for
construction and calibration of a groundwater model. By assuming the relationship to be spatially
dependent can account for two challenges: i) there may be actual changes in the petrophysical
relationship within an investigated domain, and ii) there may be resolution limitations in the estimated
resistivity model.
Challenge i) can for example be expected for sedimentary environments, where the formation
resistivity is primarily controlled by the pore water resistivity and the clay content. In the case of
spatially changes of pore water resistivity and/or content of various clay minerals content the
discrimination between clay and sands may be less clear in the estimated resistivity values. For large-
scale groundwater system the variation of pore water resistivity (e.g. saline pore water) is expected to
vary smoothly, which would be accounted for by the spatially varying petrophysical relationship.
However, the procedure only works if the underlying assumption that clay rich deposits have lower
electrical resistivity compared to sands deposits is valid.
Challenge ii) concerns the geophysical model resolution of the true formation resistivity. EM methods
are, by nature, more sensitive to deposits of low electrical resistivity than to deposits of high
resistivity, and their vertical and horizontal resolution decrease with depth. This challenge is what
affects the resistivity models estimated in the present synthetic study. Estimating spatially dependent
shape factors by groundwater model calibration let them take a compensatory role for the resolution
issues of the estimated geophysical voxel model. The calibrated shape factors may no longer have firm
physical meaning since they mainly act as correction parameters for absorbing structural errors of the
geophysical model. The estimation of locally unreasonable shape factors may be acceptable as long as
the resulting hydraulic conductivity values are reasonable. The idea of calibrating the shape factors is
related to the concept of compensatory parameters in highly parameterized calibration described by
Doherty and Welter (2010) and by Doherty and Christensen (2011).
Finally, Auken et al. (2008) showed that using borehole data as a priori information in the geophysical
inversion improves the reconstruction of the model features significantly. Estimation of EM-based
resistivity models should therefore in general be supported  by borehole information to improve the
decreasing spatial resolution of the EM methods.





## 5.2 Geophysical inversion strategy and Data quality

Inversion of AEM data using a 1D geophysical model usually applies smoothness constraints in order to regularize the inversion (Auken and Christiansen 2004; Viezzoli et al. 2008). Traditionally, the regularization includes both lateral and vertical smoothing constraints (Constable et al. 1987) or a few layer parametrization (Auken et al. 2008). Inversion using the former type of regularization produces smooth images with blurred formation boundaries which can be problematic when it is important to resolve structural connections in a complex geological system. The latter few-layer inversion may is also be prone to produce artifacts when used to map complex geological environments. It has therefore been recognized, e.g. by Day-Lewis (2005) and others, that the regularization used to stabilize the geophysical inversion may lead to artifacts that do not reflect the actual hydrogeological conditions. Thoughtless use of such results to construct groundwater models for making hydrologic predictions can therefore have serious ramifications.

Furthermore, for the present case study, the number of vertical transitions is a great challenge for the AEM method due to the principle of high resistivity equivalence: that is, it is difficult to resolve a high-resistivity layer between two low-resistivity layers because the energy loss, and therefore the sensitivity is concentrated in the more resistive layers. This will result in layer suppression, because the data sensitivity to the high resistive layer is low (Christiansen et al. 2006). This effect is present for both the smooth and sharp inversion, but in the sharp inversion the effect is less fuzzy and features, especially for the fifth layer, are more clearly reconstructed (Figure 4). When the sensitivity of the AEM method is too low, the contribution from the regularization may dominate, and information might migrate from areas with higher measurement sensitivity (Vignoli et al. 2015). In contrast to the smooth regularization scheme, the sharp regularization method is designed to penalize smooth transitions, which eventually improves the reconstruction of the deeper sand bodies. Therefore, for the present case study the sharp regularization methodology should be preferred over smooth regularization, because sharp constraints correspond better to the true structures of the reference system (categorical deposits with sharp transitions; Figure 4). Moreover, because the sharp regularization methodology leads to improved reconstruction of subsurface structures, these models lead to greater accuracy and improvement of most groundwater model predictions (Figure 9).

The groundwater system considered here is relatively shallow, at least seen from the perspective of the AEM system used in the demonstration example. This is evident from the transmitted EM signal (Figure 3). The background noise is primarily affecting the last time-gates ($10^{-4}$-$10^{-3}$s) of the low-moment and only f to a small degree the high moment time gates (even for low quality data). This implies that the resolution of the AEM data is generally high for the upper layers. Therefore, in the





present case the upper layers of all the geophysical models (both SHI-smooth and SHI-sharp) are well-
resolved and to a large extent unaffected by AEM data quality (Figure 5). However, the deep sand
units are difficult to resolve because they give only a weak signature in the AEM data (Figure 3,
Figure 5). This is particularly true for the poorest AEM data quality cases where the late time gates for
the low moment measurements are disturbed by background noise.

## 6   Summary and Conclusion

We present a workflow for automated construction and calibration of large-scale groundwater models
using a combination of airborne electromagnetic (AEM) data and hydrological data, but other types of
data could be integrated as well. First the AEM data are inverted to form a 3D geophysical model.
Subsequently, the geophysical model is translated to a 3D model of hydraulic conductivity by using a
spatially dependent petrophysical relationship for which the shape parameters are estimated by fitting
the groundwater model to hydrological data. The estimated shape factors of the petrophysical
relationship primarily work as translators between resistivity and hydraulic conductivity, but they can
also compensate for structural defects in the model.
The method is demonstrated for a synthetic case study where the subsurface consists of categorical
deposits with different geophysical and hydraulic properties. The AEM data are inverted using both
smooth and sharp regularization constraints, resulting in two competitive geophysical models.
Furthermore, the influence of the AEM data quality is tested by inverting the sharp geophysical
models using data corrupted with four different levels of background noise. The resulting groundwater
models are each calibrated on basis of head and discharge data, and their predictive performance is
tested for four types of prediction beyond the calibration base. Predictions of a pumping well's
recharge area and groundwater age are applying the same stress situation as applied during hydrologic
model calibration, while predictions of head and stream discharge is done for a changed stress
situation.
It is found that a geophysical model inverted with sharp constraints (SHI-sharp) leads to a more
accurate groundwater model than one that is based on a geophysical model inverted with smooth
constraints (SHI-smooth). The SHI-sharp model leads to an estimated hydraulic conductivity field of
greater accuracy and to improvement of most groundwater model predictions. The explanation is that
the reference system (like many real hydrogeologic systems) is characterized by sharp transitions
between the categorical deposits; this is resolved better by the SHI-sharp model than by the SHI-
smooth model.



1 Finally, it is shown that prediction accuracy improves with AEM data quality for predictions of

2 recharge area, head change and stream discharge, while the accuracy appears to not improve for

3 prediction of groundwater age.



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



Table 1. Model structural accuracy comparison for groundwater model using both smooth or sharp
geophysical models and different background noise levels. The results are averaged over the 20
system realizations. A value of 1.0 means that the model's hydraulic conductivity field is in good
agreement with the reference field; a value of 0.0 means no agreement (see body text for exact
definition of "structural accuracy").

|  | Layer 1 | Layer 2 | Layer 3 | Layer 4 | Layer 5 |
|---|---|---|---|---|---|
| SHI-1 Smooth | 0.89 | 0.79 | 0.56 | 0.54 | 0.64 |
| SHI-1 Sharp | 0.96 | 0.91 | 0.81 | 0.61 | 0.48 |
| SHI-3 Sharp | 0.96 | 0.92 | 0.82 | 0.64 | 0.5 |
| SHI-5 Sharp | 0.96 | 0.91 | 0.78 | 0.64 | 0.49 |
| SHI-10 Sharp | 0.96 | 0.9 | 0.78 | 0.6 | 0.46 |





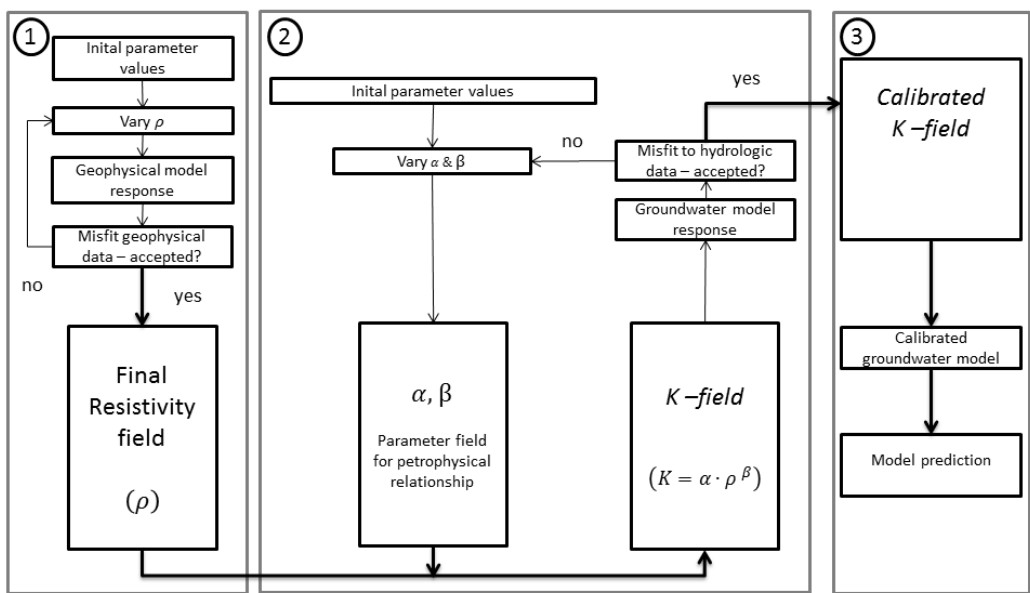

Figure 1. Conceptual flowchart for the sequential hydrogeophysical inversion. First step (box 1):
geophysical inversion. Second step (box 2), groundwater model calibration where shape factors of the
petrophysical relationship is estimated using hydrological data. Third step (box 3): The calibrated
groundwater model is used for predictive modeling.





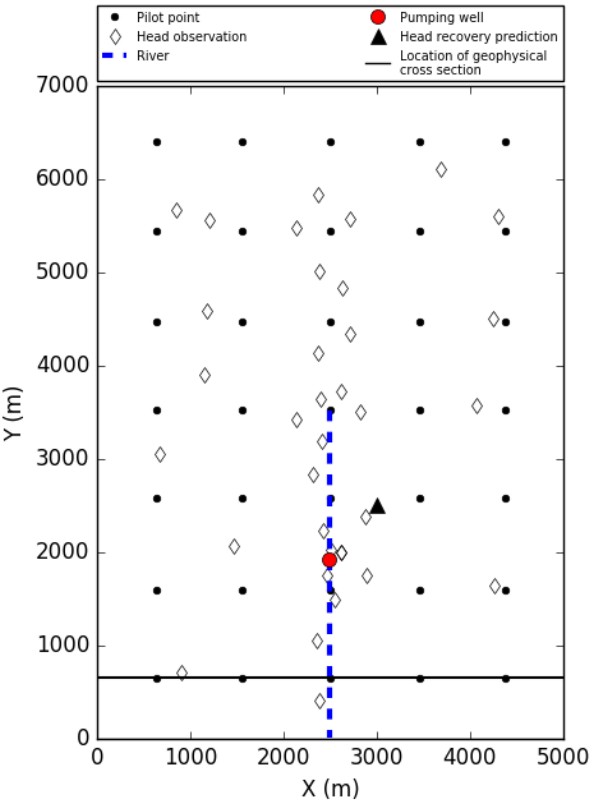

2    Figure 2. A map of locations of boreholes, a pumping well, pilot points, head recovery prediction and

3    location of a geophysical cross-section.





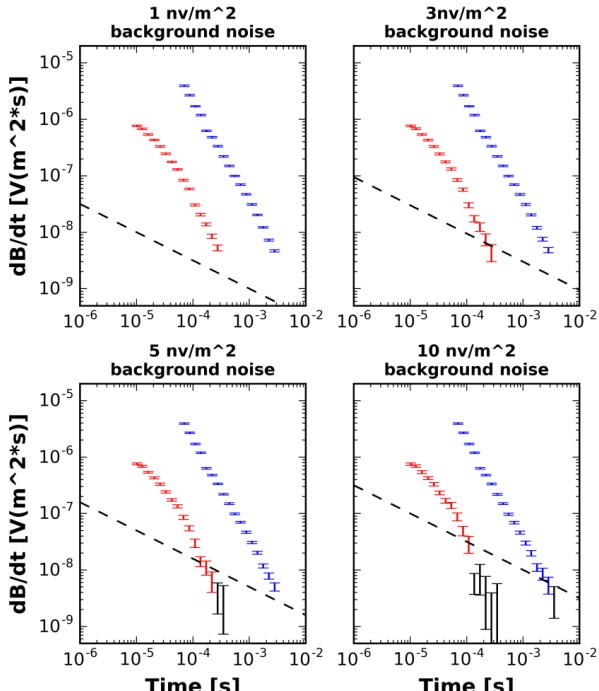

Figure 3. AEM sounding data corrupted by four levels of background noise. The black dashed curves
indicate the background noise levels, low and high moment earth responses are illustrated as red and
blue error bars, respectively, and the black error bars illustrate data which are removed by the data
processing





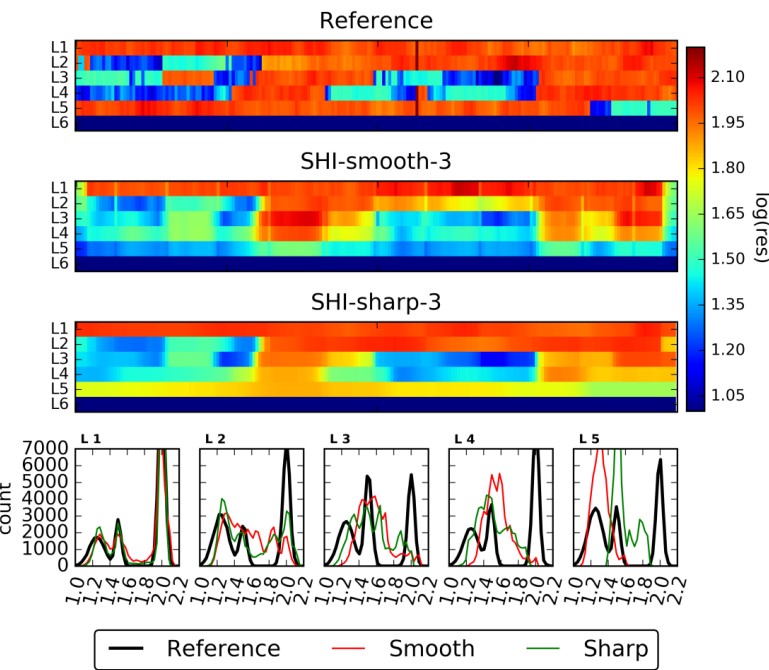

Figure 4. The figure shows an East-West cross section of resistivity for the reference system
(realization number 20), and inversion results for Smooth and Sharp inversion, respectively. The last
row shows at histogram of resistivity for each layer. The black curve is the resistivity distribution for
the reference system, the red curve shows the resistivity distribution for the smooth inversion, and
finally the green curve shows the resistivity distribution for the smooth inversion.



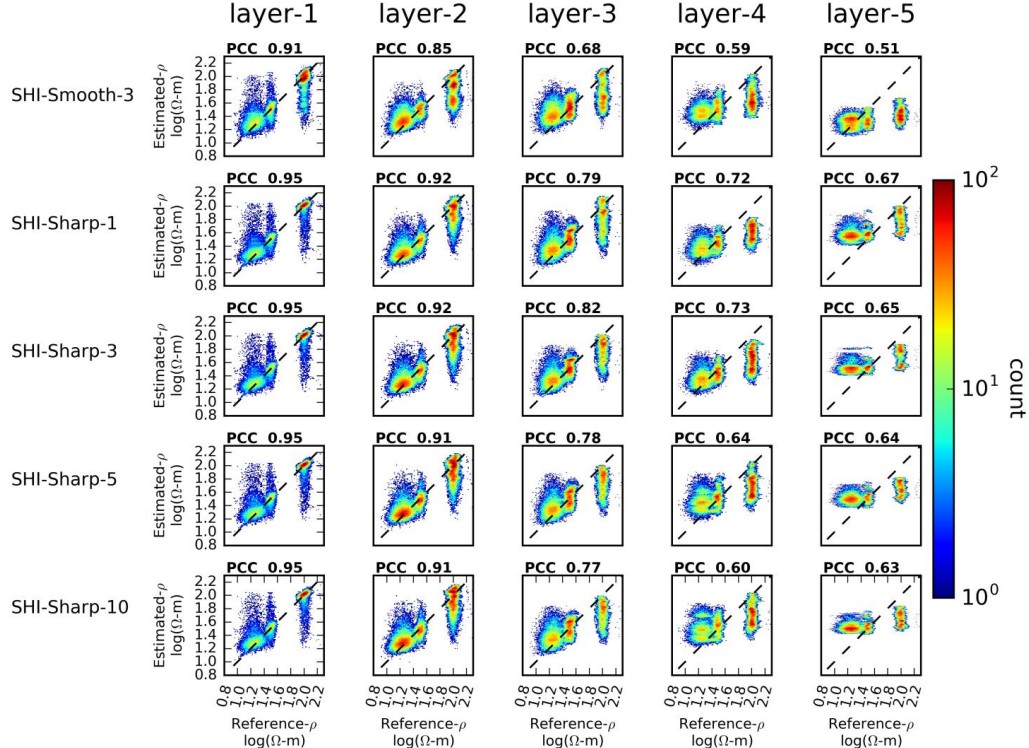

2    Figure 5. Scatterplot of true and estimated electrical resistivity field for smooth geophysical inversion

3    and sharp geophysical inversion for different data quality of the AEM data for model realization

4    number 20. On top of each window is Pearson correlation coefficient (PCC) calculated.



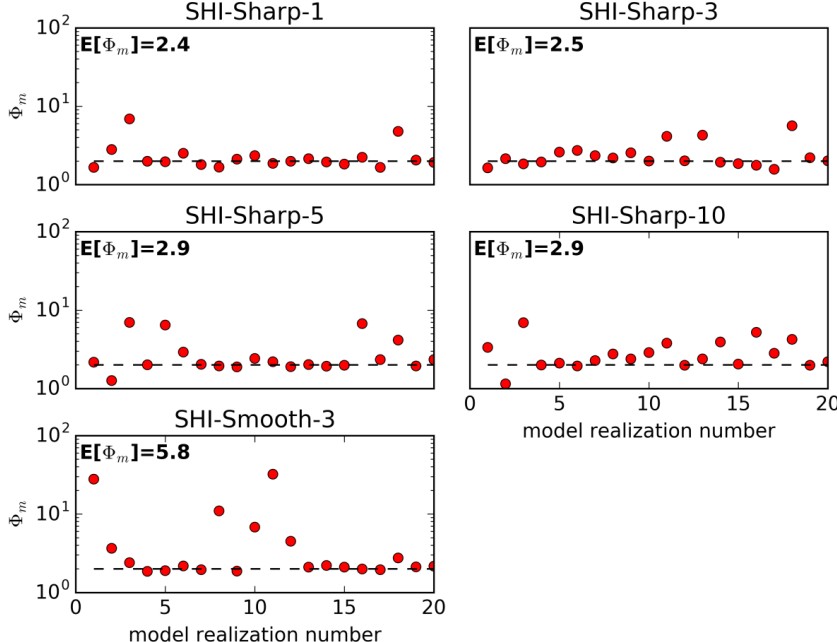

2 Figure 6. Measurement objective function value obtained for the various groundwater model

3 calibration cases, while $E[\Phi_m]$ is the mean value across all 20 different system realizations. The

4 dashed line indicates the expected target value for the model calibrations.





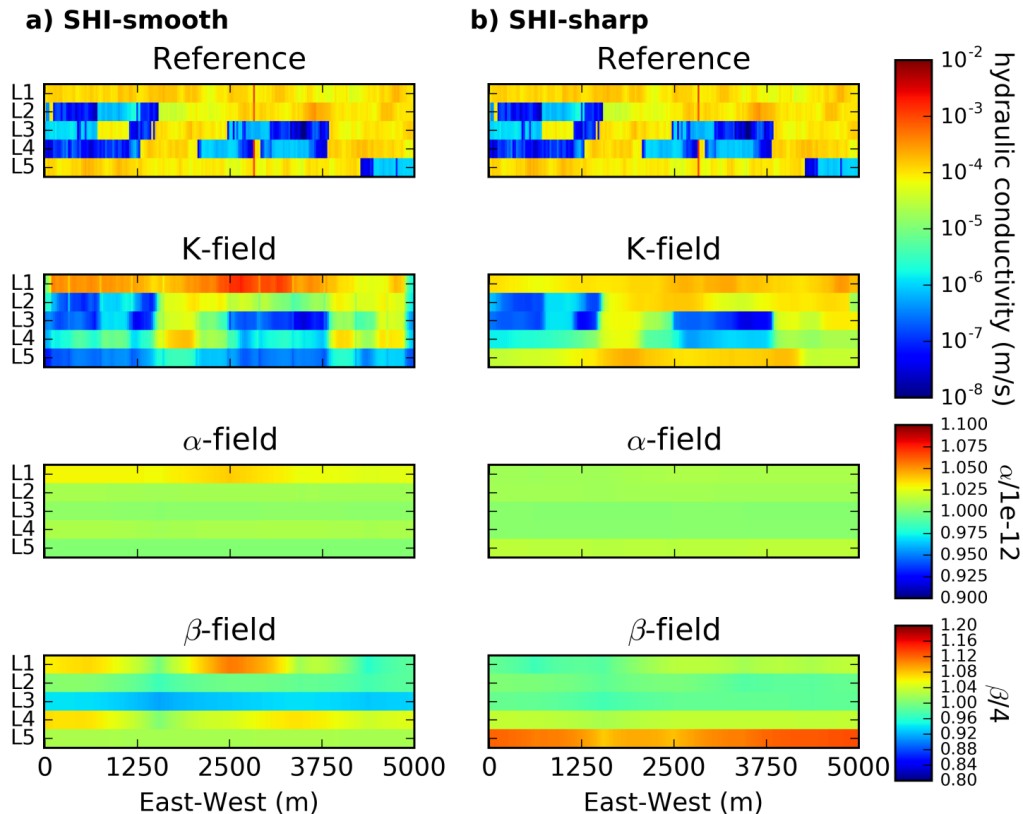

Figure 7. East-West cross-section for model realization number 20. a) shows the parameters fields for
the SHI-smooth calibrated model. b) Shows the parameters fields for the SHI-sharp calibrated model.
First row shows the reference K-field, second row shows the estimated K-field, third and fourth row
shows shape factors of the petrophysical relationship for alfa and beta, respectively.





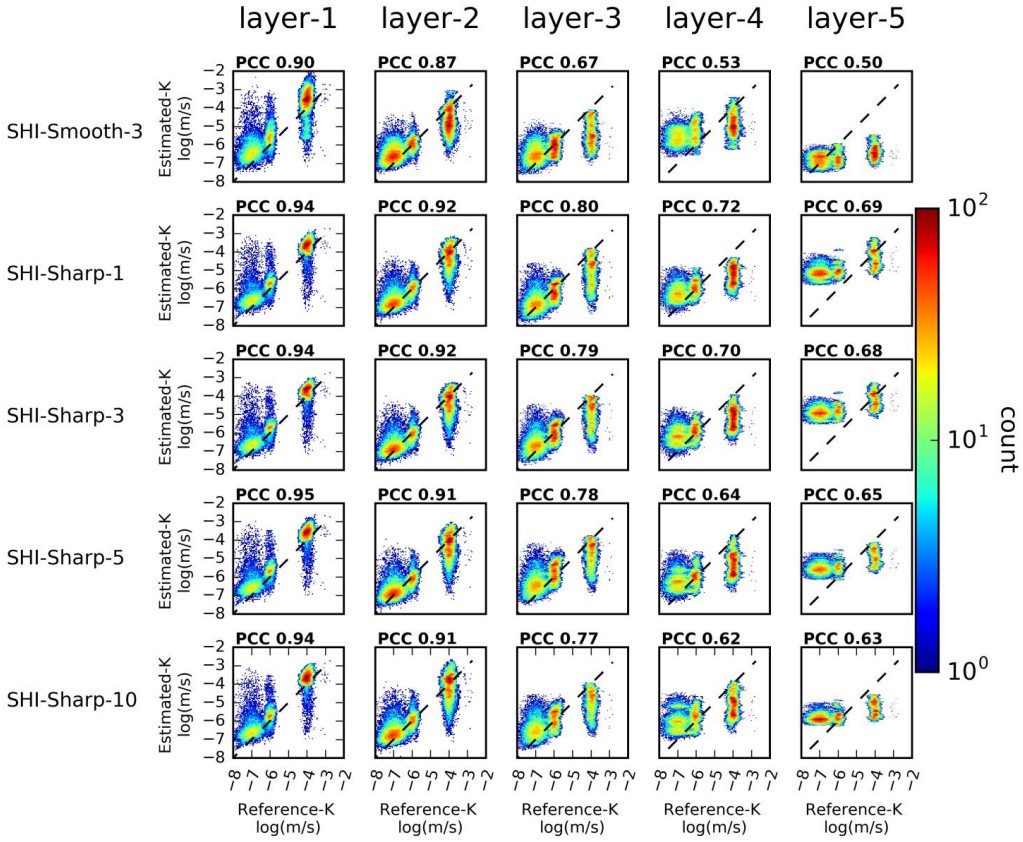

2  Figure 8. Scatterplot of true and estimated hydraulic conductivity field for smooth geophysical

3  inversion and sharp geophysical inversion for different data quality of the AEM data for model

4  realization number 20. On top of each window is Pearson correlation coefficient (PCC) calculated.





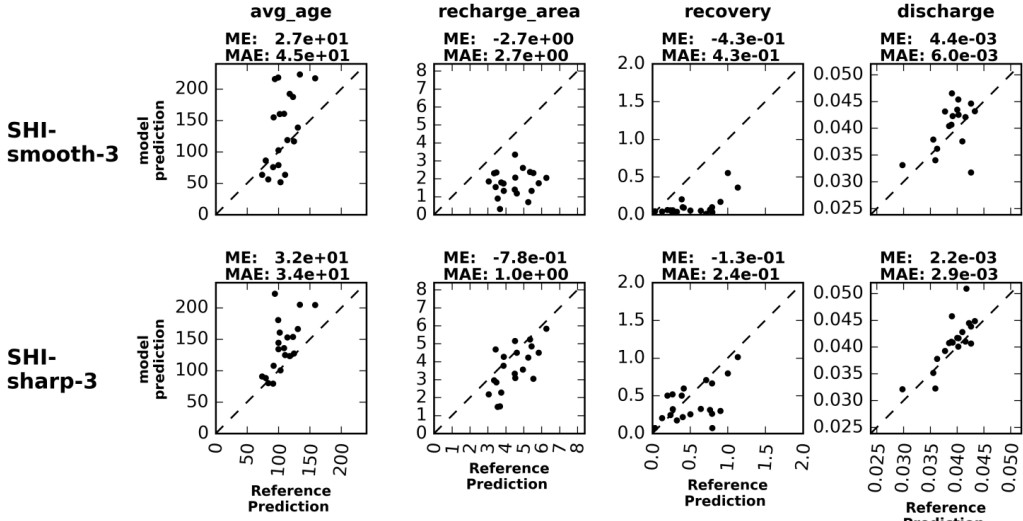

Figure 9. Scatter plots of calibrated model prediction versus the reference model prediction using
results from the 20 system realizations. The plots in the first column is for head, the second column is
for head recovery when pumping has stopped, third column is groundwater discharge to the river after
pumping has stopped, fourth and fifth column is the average age and recharge area to the pumping
well. *ME* and *MAE* are used to quantify the prediction error on basis of the 20 realizations.





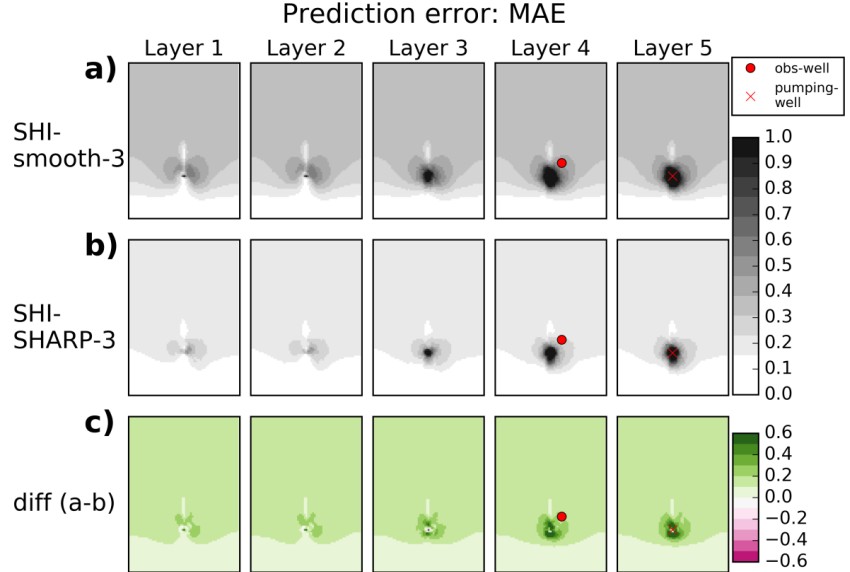

Figure 10. *MAE* contour map for head recovery prediction. a) For predictions using the SHI-smooth

models. b) For predictions using the SHI-smooth models. c) Difference between maps shown in a)

and b). Red dot marks the location of the observation well for the head recovery prediction shown in

Figure 9. The red cross marks the location of the pumping well.





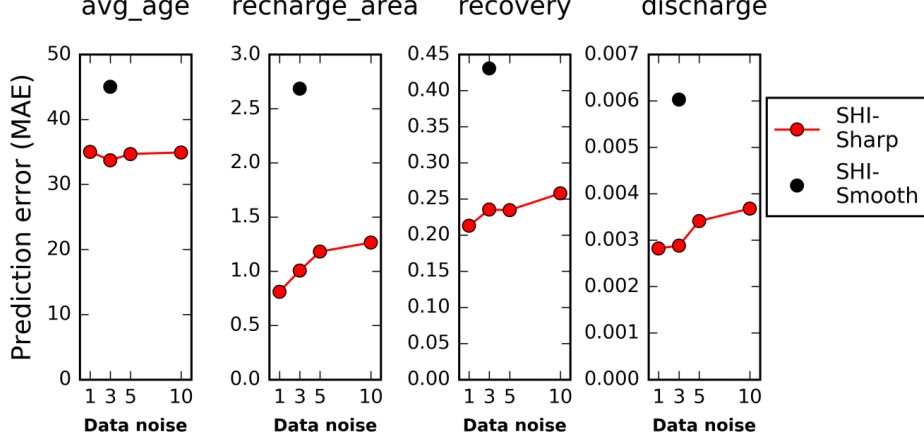

2    Figure 11. Prediction error as function of the background noise on the geophysical data. The black dot

3    is the SHI-smooth models using a background noise level of 3nV/m$^2$. The red dots are the SHI-sharp

4    models as a function of background noise level.