# Peer review of "Voxel inversion of airborne electromagnetic data for improved groundwater model construction and prediction accuracy"

_Hydrology and Earth System Sciences, 2016_

## Referee Comment (RC1) · Anonymous Referee #1 · 4 Aug 2016

Overview This study established a framework to construct and calibrate the large-scale groundwater model using AEM data as supplement rather than using only hydrological data. First, voxel inversion approach was applied to acquire a 3D geophysical model (resulting in 3D resistivity field) using AEM data which resolved the spatial scale mismatch between traditional 1D measurement position and 3D groundwater model. Then, two shape parameters of petrophysical relationship which connect the resistivity filed and hydraulic conductivity field were inversed to produce spatial optimal 3D hydraulic conductivity. Final, the calibrated hydraulic conductivity field was implemented in groundwater model to get improved model predictions. The study compared the impact of smooth constraint and sharp constraint method used in geophysical inversion

part on four types of predictions of the groundwater model, as well as the quality of AEM data. Both the experiments design and manuscript are presented meticulously and thoroughly. However there are still some questions and comments listed below.

General comments 1. In section 3.3, depth and direction dependent horizontal constraint factors were used for both smooth and sharp inversions, and the constraint factors assigned for the two inversion methods are different. However, in the results part, the author compared the impact of the two methods on the predictions of flow model, is the comparison fair?

2. In section 3.4, the author weighted the river discharge observation more than hydraulic head observation when defined the objective function. Why the author think that the calibrated models have error in their simulation of hydraulic head but not in simulation of river discharge?

3. Figure 1 in this manuscript described the conceptual flowchart for the sequential hydrogeophysical inversion. The whole framework of the experiment process was clearly displayed by the flowchart, however the content and details of each experiment step are obscure. It is hard to understand that what kind of experiment was conducted exactly in this research without reading the text description, thus I suggest the author modify the flowchart to make it intelligible.

---

## Referee Comment (RC2) · Anonymous Referee #2 · 8 Aug 2016

This paper presents an interesting work on constructing workflow for calibration of conductivity of large scale groundwater models. Besides the traditional data used for structural mapping such as lithological logs, hydrological data and so on, the model also include AEM data to provide additional information for the model parameter estimation to improve the accuracy. The whole calibration process includes two parts: Obtaining the resistivity field according to the AEM data; Estimating the empirical shape factor with assumption of power law relationship between electrical resistivity andhydraulic conductivity. Simulation results are both reasonable and meaningful. However, there are some questions and comments about this paper:

1. This paper uses geophysical "voxel inversion" to do resistivity field estimation, and

linked the resistivity field with hydraulic conductivity field through power law. Those methods are already proposed and utilized in the past. Please highlight the new theoretical development and findings.

2. In the numerical part, all the simulations are done with pre-defined true/reference model without the realistic field data. It will be better to prove the idea with realistic field data than the synthetic model.

3. In the section 3.3, how do you get those values of constraint factors?

4. In the section 3.4, the choices of weights for head and discharge data are significantly different. Why it has such a big difference? In the reality, how could you get the weight based on "trial and error" method?

5. In the simulation part, the only case used Smooth regularization is Smooth-3. What is the simulation results looks like for other noise level?

---

## Author Comment (AC1) · 7 Sep 2016

We would like to thank Anonymous Referee #1 for his valuable and relevant comments. Our replies are found below.

General comments 1. In section 3.3, depth and direction dependent horizontal constraint factors were used for both smooth and sharp inversions, and the constraint factors assigned for the two inversion methods are different. However, in the results part, the author compared the impact of the two methods on the predictions of flow model, is the comparison fair?

First we determined the constrain values for one smooth model. As explained in the

manuscript no vertical constrains were applied (which we think is fine) considering the small number of layers and the shallow discretization. Normally, the vertical discretization is characterized by logarithmically increasing layer thicknesses. As explained in the manuscript we choose to work with the same model discretization for both the geophysical and hydrological model to avoid numerical discretization errors. So, to account for the fixed layer thicknesses in the geophysical model, the horizontal constrain factors was set to decrease linearly with depth (tighter bands for the deeper layers). Furthermore, the strength given to the horizontal constraints is based on experience, keeping in mind that the constrains must not be too strong preventing fitting the data. Furthermore we visually inspecting the inverted model and found (strong) inversion artifacts perpendicular to the flight lines when using the same uniform constraint factors along the flight lines as to perpendicular to the flight lines. This is a result of having more data along the flight lines compared to perpendicular to the flight lines, and why the horizontal contains is different for the two directions. The same conceptions were applied for the "sharp" inversion. We were running the sharp inversion (for the same model) with a couple of different settings (again, based on experience and in all cases fair values) and choose settings that were producing sharp structures that looked fair (without the reference system in mind, of cause). The usefulness of the resulting geophysical inversion models depends critically on an optimal choice of the vertical and horizontal regularization of the inversion. Set the constraints too tight, and the resulting models will become overly smooth and potential resolution is lost. Set the constraints too loose, and spurious model details will appear that have no bearing on the hydrogeology. Furthermore, we don't use any model analysis to weight the geophysical inversion results into the hydrological estimations. The constrain values (in all cases fair values) affects only the final geophysical models.

General comments 2. In section 3.4, the author weighted the river discharge observation more than hydraulic head observation when defined the objective function. Why the author think that the calibrated models have error in their simulation of hydraulic head but not in simulation of river discharge?

If the model is expected to not have structural defects then it would be ideal to choose the weights $\omega\_h=\sigma\_h^{(-1)}$ and $\omega\_r=\sigma\_r^{(-1)}$. However, in this case (as in all real cases) the model has structural errors that make misfit between hydraulic head data and equivalent simulated values much larger than what can be explained by observation error ($\sigma\_h$). Using $\omega\_h=\sigma\_h^{(-1)}$ will therefore cause overfitting the head data because the head misfits (the residuals) are contaminated by structural error. Residual analysis and a few experiments were therefore made (as explained in the manuscript) to show that the choice ãĂŰ$\omega\_h=(10\cdot\sigma\_h$ )ãĂŮˆ(-2) is in agreement with the magnitude of the total head error (which is the sum of observation error and structural error). Hereby we avoid overfitting the head data. As explained in the manuscript, in this case simulation of river discharge does not appear to be contaminated by the structural errors of the model, so for the discharge data we used the normally preferred weight $\omega\_r=\sigma\_r^{(-1)}$. The way we chose the weights are in agreement with common groundwater modeling practice of using residual analysis for this purpose, see for example: Christensen, S., K.R. Rasmussen & K. Møller (1998): Prediction of regional ground-water flow to streams. Ground Water, vol. 36, no. 2, p. 351-360. Christensen, S. (1997): On the strategy of estimating regional-scale transmissivity fields. Ground Water, vol.35, no. 1, p. 131-139. We can add a few more sentences about our choice of weights to the manuscript if this is recommended.

General comments 3. Figure 1 in this manuscript described the conceptual flowchart for the sequential hydrogeophysical inversion. The whole framework of the experiment process was clearly displayed by the flowchart, however the content and details of each experiment step are obscure. It is hard to understand that what kind of experiment was conducted exactly in this research without reading the text description, thus I suggest the author modify the flowchart to make it intelligible.

We don't understand? However, some references to the flowchart in the body text of the manuscript could be clearer and consist with the numbering!

---

## Author Comment (AC2) · 7 Sep 2016

We would like to thank Anonymous Referee #2 for his valuable and relevant comments. Our replies are found below.

General comments 1. This paper uses geophysical "voxel inversion" to do resistivity field estimation, and linked the resistivity field with hydraulic conductivity field through power law. Those methods are already proposed and utilized in the past. Please highlight the new theoretical development and findings.

This paper is the first to demonstrate application of voxel inversion results directly in a groundwater modeling context. Furthermore, it presents and demonstrates a novel

parameterization method for a groundwater model for which the calibration is supported by the 3D geophysical voxel model. Finally, it demonstrates the importance of choosing a geologically plausible regularization when the geophysical model is to be used in a groundwater modeling context. Furthermore, it should also be pointed out that previous studies (linking resistivity field with hydraulic conductivity field through power law) cited deals with interpretation of tomographic data that provide a high degree of resolution, thereby allowing for interpretation of spatial variability in petrophysical relationships. In large scale applications (ten to thousands of square kilometers), this type of data is generally not available at this scale.

General comments 2. In the numerical part, all the simulations are done with pre-defined true/reference model without the realistic field data. It will be better to prove the idea with realistic field data than the synthetic model. We disagree with the referees saying that "it will be better to prove the idea with realistic field data than the synthetic model". Nothing can be "proved" from a real field case using real data; this can only be used to "demonstrate" that the method can be applied in practice and that it can produce results that appear to be plausible. The results from a real field case can only be evaluated by subjective plausibility.

This fact is actually our reason for using a synthetic model with "realistic complexity" and "synthetic data sets" that are comparable to typical data sets for a real field case. Using the synthetic case makes us able to compare model estimation results and predictions with "true fields" and "true values of the predictions". By using the synthetic case we can quantify actual estimation errors and actual prediction errors; we can for example quantify the improvement obtained by using sharp instead of smooth inversion.

Furthermore, in this case, we have tried to faithfully represent the standard practice of hydrologists in constructing models (first handling the geophysical data, hereafter the geophysical models are used as input to the hydrological construction/calibration)

General comments 3. In the section 3.3, how do you get those values of constraint factors?

This answer has also been given to referee #1: First we determined the constrain values for one smooth model. As explained in the manuscript no vertical constrains were applied (which we think is fine) considering the small number of layers and the shallow discretization. Normally, the vertical discretization is characterized by logarithmically increasing layer thicknesses. As explained in the manuscript we choose to work with the same model discretization for both the geophysical and hydrological model to avoid numerical discretization errors. So, to account for the fixed layer thicknesses in the geophysical model, the horizontal constrain factors was set to decrease linearly with depth (tighter bands for the deeper layers). Furthermore, the strength given to the horizontal constraints is based on experience, keeping in mind that the constrains must not be too strong preventing fitting the data. Furthermore we visually inspecting the inverted model and found (strong) inversion artifacts perpendicular to the flight lines when using the same uniform constraint factors along the flight lines as to perpendicular to the flight lines. This is a result of having more data along the flight lines compared to perpendicular to the flight lines, and why the horizontal contains is different for the two directions.

The same conceptions were applied for the "sharp" inversion. We were running the sharp inversion (for the same model) with a couple of different settings (again, based on experience and in all cases fair values) and choose settings that were producing sharp structures that looked fair (without the reference system in mind, of cause).

The usefulness of the resulting geophysical inversion models depends critically on an optimal choice of the vertical and horizontal regularization of the inversion. Set the constraints too tight, and the resulting models will become overly smooth and potential resolution is lost. Set the constraints too loose, and spurious model details will appear that have no bearing on the hydrogeology. Furthermore, we don't use any model analysis to weight the geophysical inversion results into the hydrological estimations. The
constrain values (in all cases fair values) affects only the final geophysical models.

The way we chose the constraint factors are in agreement with common geophysical modeling practice, see for example: Sharp spatially constrained inversion with applications to transient electromagnetic data, Geophysical Prospecting, 63, 1, 243-255. 2015, Vignoli, G., G. Fiandaca, A. V. Christiansen, C. Kirkegaard, and E. Auken

A comparison of helicopter-borne electromagnetic systems for hydrogeologic studies, Geophysical Prospecting, 2015, 1-24. 2015, Bedrosian., P., C. Schamper, and E. Auken.

General comments 4. In the section 3.4, the choices of weights for head and discharge data are significantly different. Why it has such a big difference? In the reality, how could you get the weight based on "trial and error" method?

This answer has also been given to referee #1: If the model is expected to not have structural defects then it would be ideal to choose the weights $\omega\_h=\sigma\_h^{(-1)}$ and $\omega\_r=\sigma\_r^{(-1)}$. However, in this case (as in all real cases) the model has structural errors that make misfit between hydraulic head data and equivalent simulated values much larger than what can be explained by observation error ($\sigma\_h$). Using $\omega\_h=\sigma\_h^{(-1)}$ will therefore cause overfitting the head data because the head misfits (the residuals) are contaminated by structural error. Residual analysis and a few experiments were therefore made (as explained in the manuscript) to show that the choice ãĂŰ$\omega\_h=(10\cdot\sigma\_h$)ãĂŮ^(-2) is in agreement with the magnitude of the total head error (which is the sum of observation error and structural error). Hereby we avoid overfitting the head data. As explained in the manuscript, in this case simulation of river discharge does not appear to be contaminated by the structural errors of the model, so for the discharge data we used the normally preferred weight $\omega\_r=\sigma\_r^{(-1)}$. The way we chose the weights are in agreement with common groundwater modeling practice of using residual analysis for this purpose, see for example: Christensen, S., K.R. Rasmussen & K. Møller (1998): Prediction of regional ground-water flow to streams. Ground Water, vol. 36, no. 2, p.

351-360. Christensen, S. (1997): On the strategy of estimating regional-scale transmissivity fields. Ground Water, vol.35, no. 1, p. 131-139. We can add a few more sentences about our choice of weights to the manuscript if this is recommended.

General comments 5. In the simulation part, the only case used Smooth regularization is Smooth-3. What is the simulation results looks like for other noise level?

Good question! We did not analyze other smooth models than "smooth-3", because when we saw the "smooth-3" and "sharp-3" results it convinced us that for the studied case the smooth model will always perform worse than the sharp model. This is because the geology of the synthetic system consists of "large-scale" structures of categorical fields with sharp transitions (like in a North-European or North-American glacial landscape). "Smooth inversion" cannot produce sharp transitions, so it is unlikely that a "smooth model" can do as good as a "sharp model". We therefore only use the one "smooth-3" example to demonstrate ramifications of using smooth instead of sharp inervsion. We do not see value in performing the comparison for other noise levels. Furthermore, doing the remaining smooth simulations would be computationally expensive (approx. 2-3 weeks using 64 CPU′s).

---

## Author Response (AR1)

First, we would like to thank the editor and the two anonymous referees for their valuable and relevant comments. Our replies are found below.

**Authors' corrections to the manuscript**

Besides answering the questions from the referees we have also made some editing to the manuscript:

We have made some editorial changes for English usage (Nothing major, just a bit of native-speaker polish.).

The Abstract and parts of the Introduction have been rewritten to pinpoint the manuscript's contributions.

Page 8, line 26 and line 27. The reference of Christensen et al (2015) was an old HESSD reference. The manuscript has now been published and is therefore cited as Christensen et al (2016).

Page 17 line 21-22: To clarify, " On the contrary, SHI models…" has been changed to " On the contrary, SHI-sharp models…"

Page 19 line 3: We have corrected the statement "The superiority of SHI-smooth-3 models…" to "The superiority of SHI-sharp-3 models…"

Page 21 line 16-17: We have corrected the statement "… and therefore the sensitivity is concentrated in the more resistive layers." to "… and therefore the sensitivity is concentrated in the less resistive layers."

Fig. 3. Y-label has been corrected, and some additional text has been added to the figure caption.

Fig. 7. To clarify we have updated the figure text and the caption from "SHI-smooth" to "SHI-smooth-3" and from "SHI-sharp" to "SHI-sharp-3".

Fig. 9. We have updated the text in the caption, because the order of predictions (the columns) in the figure did not correspond to the caption text.

**Referee #1: The author's response and corrections**

| Comments from Referee # 1 | Authors Comments | Changes in manuscript |
|---|---|---|
| **General comments 1.** *In section 3.3, depth and direction dependent horizontal constraint factors were used for both smooth and sharp inversions, and the constraint factors assigned for the two inversion methods are different. However, in the results part, the author compared the impact of the two methods on the predictions of flow model, is the comparison fair?* | | Yes, the reviewer is right. This section was not clear enough. We have added a few more sentences about our choice of constraint factors and how these were determined. |
| **General comments 2**. *In section 3.4, the author weighted the river discharge observation more than hydraulic head observation when defined the objective function. Why the author think that the calibrated models have error in their simulation of hydraulic head but not in simulation of river discharge?* | | We have rewritten the explanation of our choice of weights that follows after equation (7). |
| **General comments 3**. *Figure 1 in this manuscript described the conceptual flowchart for the sequential hydrogeophysical inversion. The whole framework of the experiment process was clearly displayed by the flowchart, however the content and details of each experiment step are obscure. It is hard to understand that what kind of experiment was conducted exactly in this research without reading the text description, thus I suggest the author modify the flowchart to make it intelligible.* | We don't understand? However, some references to the flowchart in the body text of the manuscript could be clearer and consistent with the numbering! | We have made a few improvements to figure 1 to clarify the workflow. Page 7 line 10: (**Error! Reference source not found.**, box 1) changed to (figure 1, box 2) |

**Referee #2: The author's response and corrections**

| Comments from referee # 2 | Authors Comments | Changes in manuscript |
|---|---|---|
| *This paper uses geophysical "voxel inversion" to do resistivity field estimation, and linked the resistivity field with hydraulic conductivity field through power law. Those methods are already proposed and utilized in the past. Please highlight the new theoretical development and findings.* | This paper is the first to demonstrate application of voxel inversion results directly in a groundwater modeling context. Furthermore, it presents and demonstrates a novel parameterization method for a groundwater model for which the calibration is supported by the 3D geophysical voxel model. Finally, it demonstrates the importance of choosing a geologically plausible regularization when the geophysical model is to be used in a groundwater modeling context. Furthermore, it should also be pointed out that previous studies (linking resistivity field with hydraulic conductivity field through a power law) that we cite deal with interpretation of tomographic data that provide a high degree of resolution, thereby allowing for interpretation of spatial variability in petrophysical relationships. In large scale applications (ten to thousands of square kilometers), this type of data will in general not be available. | We have rewritten parts of the Abstract.

On page 4 L25-28 we added this sentences in the text:
*"However, it is often ignored that geophysical data can be inverted using alternative regularization schemes, and to test whether the alternative geophysical models affect the predictive capability of a groundwater model."* |
| *In the numerical part, all the simulations are done with pre-defined true/reference model without the realistic field data. It will be better to prove the idea with realistic field data than the synthetic model.* | We disagree with the referees saying that "it will be better to prove the idea with realistic field data than the synthetic model". Nothing can be "proved" from a real field case using real data; this can only be used to "demonstrate" that the method can be applied in practice and that it can produce results that appear to be plausible. The results from a real field case can only be evaluated by subjective plausibility.
This fact is actually our reason for using a | Nothing changed in the manuscript |

| | synthetic model with "realistic complexity" and "synthetic data sets" that are comparable to typical data sets for a real field case. Using the synthetic case makes us able to compare model estimation results and predictions with "true fields" and "true values of the predictions". By using the synthetic case we can quantify actual estimation errors and actual prediction errors; we can for example quantify the improvement obtained by using sharp instead of smooth inversion.

Furthermore, in this case, we have tried to faithfully represent the standard practice of hydrologists in constructing models (first handling the geophysical data, hereafter the geophysical models are used as input to the hydrological construction/calibration) | |
|---|---|---|
| *In the section 3.3, how do you get those values of constraint factors?* | This answer has also been given to referee #1: | Yes, the reviewer is right. This section was not clear enough. We have added a few more sentences about our choice of constraint factors and how these were determined. |
| *In the section 3.4, the choices of weights for head and discharge data are significantly different. Why it has such a big difference? In the reality, how could you get the weight based on "trial and error" method?* | This answer has also been given to referee #1:

We can add a few more sentences about our choice of weights to the manuscript if this is recommended. | As said above to referee #1, we have rewritten the explanation of our choice of weights that follows after equation (7). |
| *In the simulation part, the only case used Smooth regularization is Smooth-3. What is the simulation results looks like for other noise level?* | Good question! We did not analyze other smooth models than "smooth-3", because when we saw the "smooth-3" and "sharp-3" results it convinced us that for the studied case the smooth model will always perform worse than the sharp model. This is because the geology of the synthetic system consists of | Nothing changed |

| | | |
|---|---|---|
| | "large-scale" structures of categorical fields with sharp transitions (like in a North-European or North-American glacial landscape). "Smooth inversion" cannot produce sharp transitions, so it is unlikely that a "smooth model" can do as good as a "sharp model". We therefore only use the one "smooth-3" example to demonstrate ramifications of using smooth instead of sharp inervsion. We do not see value in performing the comparison for other noise levels. Furthermore, doing the remaining smooth simulations would be computationally expensive (approx. 2-3 weeks using 64 CPU´s). | |

[revised manuscript text omitted]

---

## Author Response (AR2)

1. In the section 3.3, page 12, line 12: "The strength given to the horizontal constraints is based on experience". In both cases, the constraint factors are ~ 1. Does it mean for different applications, the constraints should also be around 1? Please clarify.

We clarified and further explained the meaning of the constraint factors both in section 2.1 and 3.3

Section 2.1

"The geophysical voxel inversion is carried out on the logarithm of the resistivity values ($m = log(\rho)$), and the constraints values are expressed in terms of constraint factors $CFs$, representing the relative strength of the constraints (Auken et al., 2014). The actual values of the constraint standard deviations $\sigma_{i,j}$ of eq. (1) and eq. (2) are then computed as $\sigma_{i,j} = log(CF_{i,j})$. For instance, a constraint factor value of $CF_{i,j} = 1.9$ gives $(m_i - m_j)^2/\sigma_{i,j}^2 = 1$ in eq. (1) when $\rho_i/\rho_j = 1.9$, i.e. when the resistivity values are 90% different (Vignoli et al. 2015)."

Section 3.3

"All the constraints values used in this study represent typical values working also in other applications, both for synthetic and filed data."

2. Page 36, Figure 4. Figure label: green curve should be "sharp inversion"

Done